# Design of a Ku-Band Monopulse Antenna with a Truncated Reflector and an Open-Ended Waveguide Feed

**DOI:** 10.3390/s23010118

**Published:** 2022-12-23

**Authors:** Ayodeji Matthew Monebi, Chan-Soo Lee, Bierng-Chearl Ahn, Seong-Gon Choi

**Affiliations:** School of Electrical and Computer Engineering, Chungbuk National University, Cheongju 28644, Republic of Korea

**Keywords:** monopulse antenna, truncated parabolic reflector, sum and difference channels

## Abstract

This paper presents a design for a monopulse reflector antenna with asymmetric beamwidths for radar applications at the Ku band. The proposed design features a rectangular waveguide monopulse feed and a truncated parabolic reflector. An array of four open-ended rectangular waveguides were employed to realize a compact monopulse feed. The reflector is cut in the *H* plane of the feed producing a wider beam in the azimuth plane. This type of pattern is useful in applications such as projectile tracking and airport surveillance. The design parameters for optimum performances are chosen at all stages of the design. The design and analysis have been carried out using the commercial simulation tool CST Studio Suite 2022. The directivity of the sum, elevation difference and azimuth difference channels of the reflector antenna are 32.1, 28.1, and 26.4 dB at 14 GHz; 30.9, 29, and 27.3 dB at 15 GHz; 31.7, 29.6, and 27.6 dB at 16 GHz; 31.6, 29.9, and 27.8 dB at 17 GHz.

## 1. Introduction

The monopulse technique can be used to determine the target’s position from a single pulse by comparing the received signals obtained concurrently from one or more antenna beams [1]. This makes it significantly better than other techniques as it delivers an unmatched accuracy of the target’s position and angle. Reflector antennas have been a popular solution for delivering high-gain communications at high frequencies in areas such as satellite communications, radar, and astronomy. 

It is worth mentioning that researchers have developed different kinds of monopulse antennas for radar applications [1,2,3,4,5]. In [6,7,8], the monopulse feed was designed by leveraging a dielectric lens. In [9], researchers investigated a monopulse reflector antenna operating at 35 GHz for missile applications. In [10], a parabolic reflector with a compact monopulse feed was developed for earth observation use. The investigation of the monopulse antenna soars in [11] included researching a W-band tracker and using a monopulse feed parabolic antenna. In [12], a monopulse beam tracking system for UAV use was reported. Investigations advanced with [13] using the monopulse feed technique to address long-range communications from the air to the ground. A matched feed illuminating parabolic reflector was investigated in [14,15,16]. In [17], antenna feed with offset parabolic reflectors was studied. The electric field distribution of the fast and slow modes in the open-ended waveguide was reported in [18]. In [19], a parabolic antenna design for coastal radar surveillance was reported using a dual horn feed.

In some applications, a monopulse antenna with a narrow beam in one plane and a wider beam in the other plane was required. However, investigations that achieved a concurrent sharp pencil beam at the elevation plane, an outspread beam at the azimuth plane, and a robust sum channel beam with minimum side lobes have not been sufficiently explored. Study [20] designed a truncated paraboloid reflector antenna, however, with a single antenna feed system.

Monopulse feed design used to be very complicated because one needs to carefully combine four or five apertures to arrive at the symmetric and balanced sum and difference patterns. In this paper, we propose a very simple monopulse feed suitable for illuminating a truncated parabolic reflector. The proposed feed utilizes four rectangular waveguide apertures that are excited by a coaxial probe. It is easier to implement a monopulse comparator in a coaxial medium than in waveguide form. We employed four rectangular waveguide open-ends, which are compact and simple to realize.

Few results have been published regarding monopulse reflector antennas with asymmetric beamwidths suitable for the tracking of flying objects. In this paper, we propose a design for an optimally illuminated truncated parabolic reflector antenna with well-balanced sum and difference patterns.

First, a design is carried out for a monopulse consisting of four open-ended waveguide radiators, and the sum and difference gain patterns of the feed are analyzed to find an optimum reflector geometry. Part of a parabolic reflector surface is removed so that the feed optimally illuminates the reflector. Essentially, the purpose of our paper is to present an idea, not to show the result of development efforts. In this regard, we have used CST Studio Suite 2022, a well-proven simulation tool whose accuracy has been confirmed in many published papers so that the simulation alone is enough to prove the proposed concept. 

## 2. Feed Design

Figure 1 shows the structure and dimensional parameters of the monopulse feed presented in this paper. The feed consists of four rectangular-waveguide open-end radiators on a circular ground plane or flange. The beauty of the proposed feed is its simplicity. The waveguide is excited by a coaxial probe. Four waveguide radiators are combined to form a sum pattern, an elevation difference pattern, and an azimuthal difference pattern that are suitable for illuminating a parabolic reflector. The waveguide aperture dimensions *a* and *b* are chosen so that the feed will give suitable sum and difference patterns at 14–17 GHz. The gap between the rectangular apertures is chosen so that the grating lobe is suppressed in the visible region of the radiation angle while the coupling between the waveguide apertures is low enough (e.g., less than −20 dB). A circular flange is employed to reduce backward radiation from the feed.

Once the feed dimensions are determined, the waveguide is fed by a coaxial probe whose diameter and position are found by iterative parametric analysis. Critical to obtaining a good impedance matching of the probe-excited rectangular monopulse feed is the probe length (*L*), the probe diameter (*E*), and the probe’s position away from the waveguide end wall (*C*). The length *L* of the probe is less than a quarter waveguide wavelength (<*λ*_g_/4), and that makes it behave similar to a capacitor, thereby introducing capacitive susceptance to match the source to the load. All the parameters are optimized for frequencies of 14–17 GHz. The cutoff frequency of the fundamental TE_10_ mode in the rectangular waveguide is 9.375 GHz, which is determined by the waveguide’s broadwall dimensions *a*. The dimensions in millimeters of the designed feed are shown in Table 1.

Figure 2 shows four open-ended waveguide apertures, where *A*_1_, *A*_2_, *A*_3_, and *A*_4_ are apertures in quadrants one, two, three, and four, respectively. A sum pattern is obtained by adding all the apertures in the phase (*A*_1_ + *A*_2_ + *A*_3_ + *A*_4_). An elevation difference pattern is obtained by deducting the two upper apertures from the two lower quadrants (*A*_1_ + *A*_2_ − *A*_3_ − *A*_4_). Apertures *A*_1_ and *A*_2_ are excited with a phase of 0° while apertures *A*_3_ and *A*_4_ are fed with a phase of 180°. This elevation difference pattern is used to compute the target’s location in the elevation plane. The azimuthal difference pattern is realized by subtracting the right vertical apertures from the left vertical ones (*A*_2_ + *A*_3_ − *A*_1_ − *A*_4_). Aperture *A*_2_ and *A*_3_ have a phase of 0°, while apertures *A*_1_ and *A*_4_ are in a 180° phase. The azimuthal difference pattern is used to obtain the target’s location in the azimuth plane.

Figure 3a,b depict the reflection coefficient and transmission coefficients between apertures. In Figure 3a, the waveguides are excited with the TE_10_ mode (wave port in CST Studio Suite), while they are excited with a coaxial probe in Figure 3b. Good results were obtained in the reflection and transmission coefficients in both cases at 14–17 GHz. Particularly, the reflection coefficient (*S*_11_) of Figure 3a shows a good return loss below 14.29 dB from 14 GHz to 17 GHz, and in Figure 3b, the reflection coefficient is less than 12 dB at 14–17 GHz. The *E*-plane coupling is still less than 15 dB at 14–17 GHz, as can be seen in the *S*_41_ graph; the *H*-plane coupling (*S*_21_) and diagonal-plane coupling (*S*_31_) are less than 31 dB at 14–17 GHz in both cases.

Figure 4a,b show the *E*- and *H*-plane directivity patterns of the sum channel. The maximum directivity of the sum channel is 13.4, 13.7, 14.3, and 14.8 dB at 14, 15, 16, and 17 GHz, respectively. The sidelobe level is less than −10.3 dB at 14–17 GHz in both *E* and *H* planes. The *E*- and *H*-plane 3-dB beamwidths are 52.2°, 29.2 ° at 14 GHz; 48.2°, 27.3° at 15 GHz; 41.6°, 25.1° at 16 GHz; and 36.3°, and 23.3° at 17 GHz. The *E*- and *H*-plane 10-dB beamwidths are 86.24°, 48.27 ° at 14 GHz; 78.65°, 45.39° at 15 GHz; 73.6°, 42.13° at 16 GHz; and 70.33°, and 39.26° at 17 GHz. 

Figure 5a shows the directivity patterns of the elevation difference channel. The result shows a directivity of 11.3 dB at 14 GHz; 11.6 dB at 15 GHz; 11.7 dB at 16 GHz; and 11.5 dB at 17 GHz. The sidelobe level is less than −19.6 dB at 14–17 GHz. The directivity of the azimuth difference channel is shown in Figure 5b. The result shows a directivity of 10.6 dB at 14 GHz; 11.4 dB at 15 GHz; 12.2 dB at 16 GHz, and 12.8 dB at 17 GHz. The sidelobe level is less than −24.8 dB at 14–17 GHz. One can observe a deep null at 0 degrees in the elevation- and azimuth-difference patterns. 

The initial phase center of the feed is obtained using the phase center functionality of the CST Studio Suite for the sum pattern. With the coordinate origin for patterns calculation at the initial phase center, the phase patterns of the sum and two different channels are calculated. An optimum phase center is obtained by adjusting the coordinate origin so that the phase variations in the 10-dB beamwidth of the sum pattern are minimum for the sum and difference channels. The optimum phase center was found to be 1.0 mm into the aperture. The *E*- and *H*-plane phase patterns of the sum channel are shown in Figure 6a,b, respectively. Figure 7a,b show the phase pattern of the elevation and azimuth difference channels, respectively. Good phase uniformity can be observed in the sum- and difference-channel phase patterns.

## 3. Reflector Design

Based on the foregoing design of the monopulse feed, a prime-focus parabolic reflector was designed. The reflector diameter was chosen for a sum gain of about 30 dB with good difference pattern characteristics. Figure 8a,b show the structure and dimensional parameters of the truncated reflector presented in this paper. The half illumination angles of the reflector in the *E*- and *H*-planes are denoted by *θ*_1_ and *θ*_2_ in Figure 8b. The equation for the reflector diameter *D*, focal length *F*, and *E*-plane illumination angle *θ*_1_ is given by Equation (1):(1)$$\theta_{1}=2\tan^{-1}\frac{1}{4F/D}$$

Since the *H*-plane beamwidth *θ*_2_ is smaller than *θ*_1_, the reflector is truncated, as shown in Figure 8a. Illumination angles *θ*_1_ and *θ*_2_ are adjusted for optimum performance in both the sum and difference patterns. When these angles are chosen for the best performance in the sum channel, the difference in the channel patterns is not good, so a compromise is required. The *θ*_1_ and *θ*_2_ parameters are obtained through an iterative calculation for the directivity pattern of the reflector. An optimum choice of *θ*_1_ and *θ*_2_ is made so that the gain requirement is met for the sum channel and a good monopulse slope performance can be obtained for the difference channels. The optimum values of *θ*_1_ and *θ*_2_ were found to be 51° and 23°, respectively. 

The feed’s phase center, at 1.0 mm inside the aperture, is placed at the focal point of the reflector. The waveguide is excited with the TE_10_ mode using a wave port in the CST Studio Suite. As before, the sum and difference patterns are obtained by the proper combinations of four apertures of the feed. The reflector and the feed are simulated together so that the effect of the feed scattering is included in the far-field directivity patterns. Table 2 shows the dimensions of the designed reflector.

Figure 9a shows the reflection coefficient of the TE_10_-mode-feed illuminating the reflector. The reflection coefficient is less than −10 dB at 14–17 GHz, albeit with a slight increase in the periodic interference effects due to the presence of the reflector. The transmission coefficient or the coupling between apertures has also been increased again with period interferences. The *E*-plane coupling is still less than −15 dB at 14–17 GHz, as seen in the *S*_41_ graph. The *H*-plane coupling (*S*_21_) and diagonal-plane coupling (*S*_31_) are less than −18 dB at 14–17 GHz. Figure 9b shows the reflection and transmission coefficients of the probe-excited feed illuminating the reflector. The reflection coefficient is slightly larger than that of the TE_10_-mode excited feed and is still less than −10 dB at 14–17 GHz. The coupling between the apertures is similar to that of the TE_10_-mode feed.

The directivity patterns of the sum and difference channels are calculated by full-wave analysis using CST Studio Suite, including the feed. The sum-channel directivity patterns at smaller angles are shown in Figure 10a,b for the *E* plane and Figure 11a,b for the *H* plane. 

The maximum directivity of the sum channel is 32.1, 30.9, 31.7, and 31.6 dB at 14–17 GHz, respectively. The elevation difference channel directivity is 28.1, 29, 29.6, and 29.9 dB at 14–17 GHz, while the azimuth difference channel directivity is 26.4, 27.3, 27.6, and 27.8 at 14–17 GHz, respectively. 

The 3-dB beamwidth of the elevation sum channel and the elevation difference channels at 14–17 GHz are obtained as follows: 2.6°, 1.8° at 14 GHz; 3.2°, 1.6° at 15 GHz; 3.0°, 1.5° at 16 GHz; and 2.8°, 1.4° at 17 GHz, respectively. The 3-dB beamwidth of the azimuthal sum channel and the azimuth difference channels at 14–17 GHz are obtained as follows: 5.2°, 3.8° at 14 GHz; 5.3°, 3.6° at 15 GHz; 5.0°, 3.4° at 16 GHz; and 4.6°, 3.3° at 17 GHz, respectively.

The sidelobe levels of the *E*-plane sum channel are −13, −20, −14.6, and −12.6 dB at 14, 15, 16, and 17 GHz, respectively. The sidelobe levels of the *H*-plane sum channel are −20, −18.8, −19.6, and −19.9 dB at the same frequencies. The sidelobe levels of the elevation difference channels are −12.6, −12, −10.4, and −9.3 dB at the same frequencies. The sidelobe levels of the azimuth difference channels are −13.3, −14, −13.7, and −13.2 dB at the same frequencies. 

The sum- and difference-channel directivity patterns at wider angles are shown in Figure 12a,b for the *E* plane and in Figure 13a,b for the *H* plane. 

Figure 14 shows the sum-channel cross-polarization patterns in the diagonal plane (*φ* = 45°). *E*- and *H*-planes are shown in the range of 14–17 GHz. The maximum directivity of the cross-polarized radiation in the *E*- and *H*-planes is 0.189, −1.50, 0.90, and 1.16 dB at 14, 15, 16, and 17 GHz, respectively. Figure 15a,b show the diagonal-plane cross-polarization directivity patterns of the elevation-and azimuth-difference channels, respectively. The maximum cross-polarization directivity of the elevation-difference channel is 1.84, 3.50, 2.50, and 4.72 dB at 14, 15, 16, and 17 GHz, respectively. The maximum cross-polarization directivity of the azimuth-difference channel is 6.42, 6.86, 6.97, and 6.47 dB at 14, 15, 16, and 17 GHz, respectively. 

Table 3 shows the comparative analysis between this work and existing results. It can be seen from Table 3 that this work achieved an asymmetrical beam performance in the range of 14–17 GHz with sharp beams in the elevation plane and spread-out azimuth beams, which are useful for applications such as projectile tracking radars, airport surveillance radars, and vehicle tracking radars.

## 4. Conclusions

The design of a monopulse antenna consisting of a simple coax-fed monopulse feed and a truncated parabolic reflector has been presented for Ku band radar applications at 14–17 GHz requiring asymmetric beamwidths in the elevation and azimuth planes. The feed is an array of four open-ended rectangular waveguides fed by a coaxial probe. The first stage of the work presents the arrangement of the radiators and the design of the coaxial probe. 

The second stage detailed a final design that combines the monopulse feed with a truncated parabolic reflector. The *E*- and *H*-plane angles of the feed illuminating the reflector have been optimized for balanced directivities of the sum and difference channels. The designed reflector antenna has the following characteristics. The co-polarization directivity of the sum channel, elevation, and azimuth difference channels are 30.9–32.1 dB, 28.1–29.9 dB, and 26.4–27.8 dB at 14–17 GHz, respectively. The cross-polarization directivity of the sum channel, elevation, and azimuth difference channels are −1.50–1.16, 1.84–4.82, and 6.42–6.97 dB at 14–17 GHz, respectively. The *E*- and *H*-plane beamwidths of the sum channel are 2.6–3.2 and 4.6–5.3 degrees at 14–17 GHz, respectively. The *E*- and *H*-plane sidelobe levels of the sum channel are −20.0 to −12.6 dB and −20.0 to −18.8 dB at 14–17 GHz. The proposed design of a monopulse reflector antenna may find applications where asymmetric beamwidths are required, such as projectile tracking radars, airport surveillance radars, and vehicle tracking radars.

## Figures and Tables

**Figure 1 sensors-23-00118-f001:**
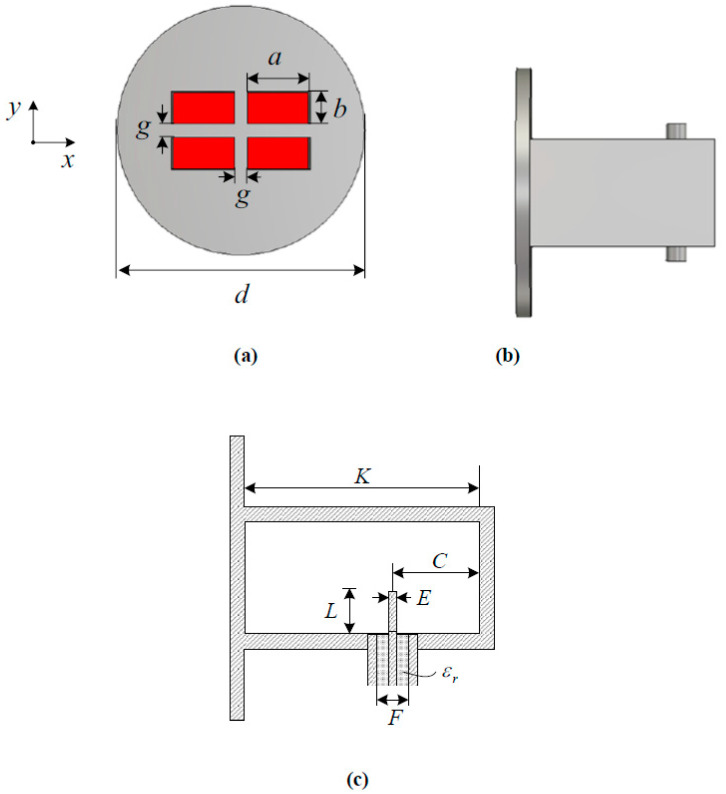
Dimensional parameters of the designed feed. (**a**) Front view. (**b**) Side view. (**c**) Probe dimensions.

**Figure 2 sensors-23-00118-f002:**
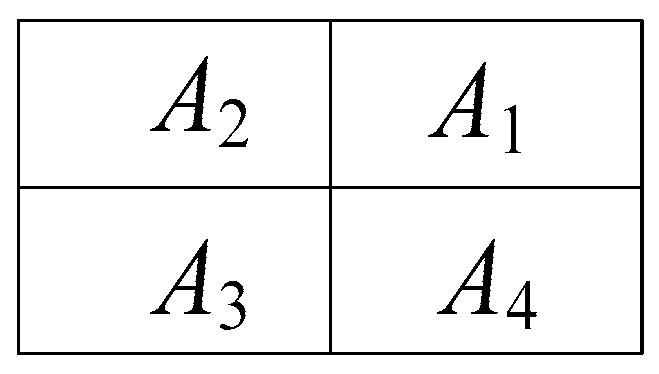
Four-quadrant aperture.

**Figure 3 sensors-23-00118-f003:**
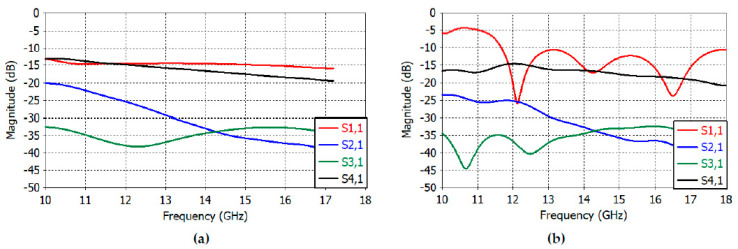
(**a**) Reflection and transmission coefficients of the TE_10_-mode-excited rectangular feed. (**b**) Reflection and transmission coefficients of the probe-excited rectangular feed.

**Figure 4 sensors-23-00118-f004:**
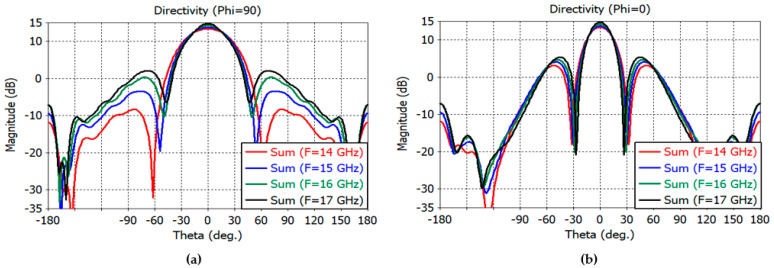
(**a**) *E*-plane directivity patterns of the sum channel. (**b**) *H*-plane directivity patterns of the sum channel.

**Figure 5 sensors-23-00118-f005:**
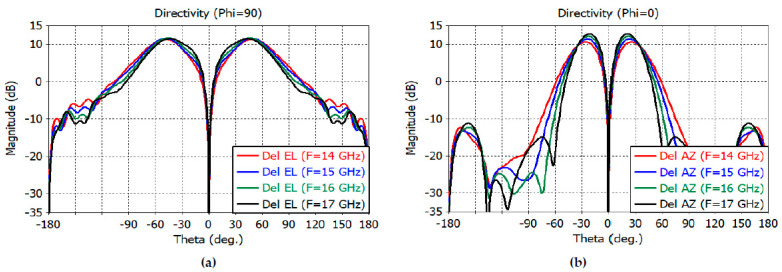
(**a**) Elevation difference channel pattern. (**b**) Azimuth difference channel pattern.

**Figure 6 sensors-23-00118-f006:**
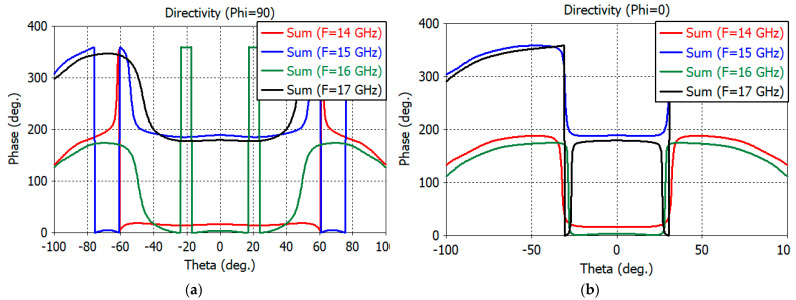
(**a**) *E*-plane phase patterns of the sum channel. (**b**) *H*-plane phase patterns of the sum channel.

**Figure 7 sensors-23-00118-f007:**
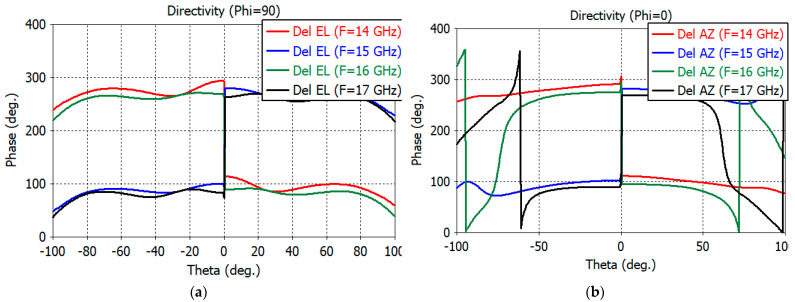
(**a**) Phase patterns of the elevation difference channel. (**b**) Phase patterns of the azimuth difference channel.

**Figure 8 sensors-23-00118-f008:**
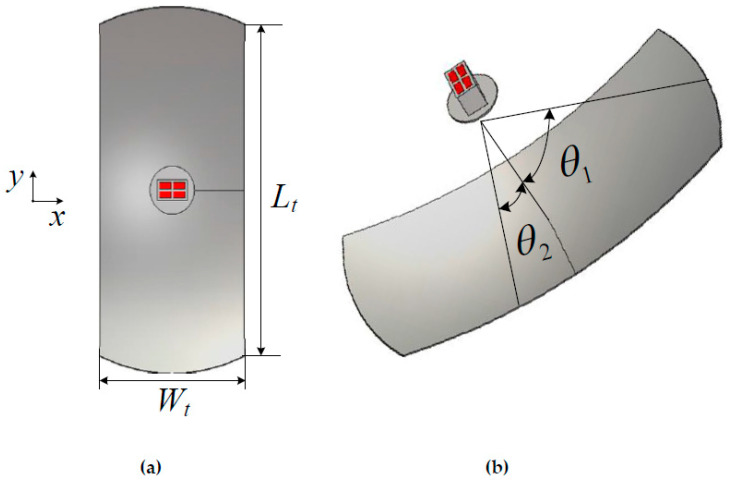
(**a**) Front view. (**b**) A 3D view of the proposed truncated parabolic reflector with a feed.

**Figure 9 sensors-23-00118-f009:**
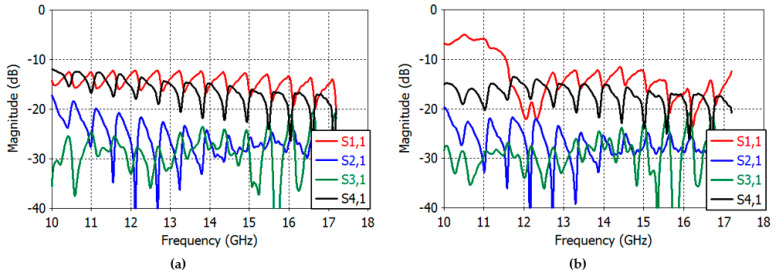
Reflection and transmission coefficients of the feed excited by TE_10_ modes (**a**) and by coaxial probes (**b**) illuminating the truncated reflector.

**Figure 10 sensors-23-00118-f010:**
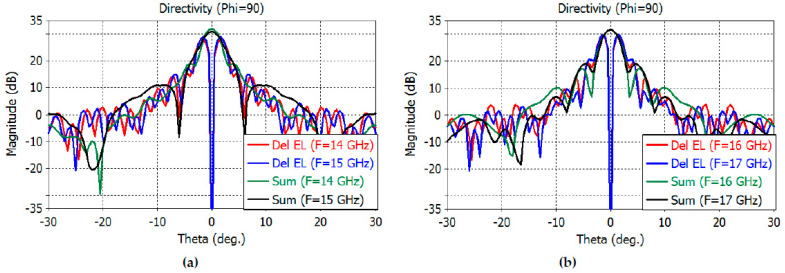
*E*-plane far-field directivity patterns of the sum channel at 14–15 GHz (**a**) and at 16–17 GHz (**b**).

**Figure 11 sensors-23-00118-f011:**
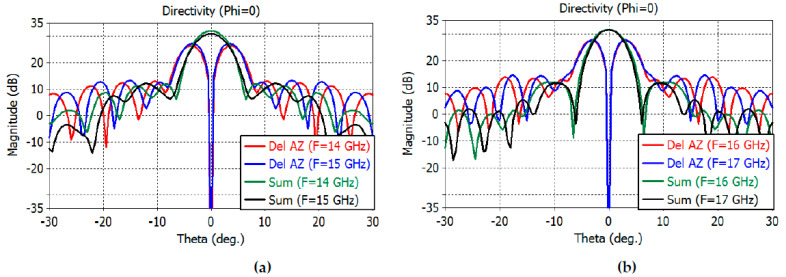
*H*-plane far-field directivity patterns of the sum channel at 14–15 GHz (**a**) and at 16–17 GHz (**b**).

**Figure 12 sensors-23-00118-f012:**
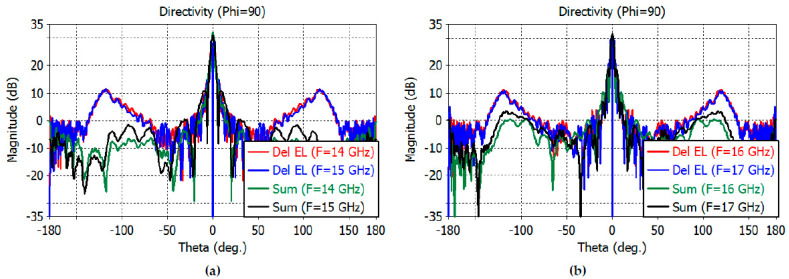
(**a**) *E*-plane directivity patterns of the reflector at 14-15 GHz (−180° to 180°). (**b**) *E*-plane directivity patterns of the reflector at 16–17 GHz (−180° to 180°).

**Figure 13 sensors-23-00118-f013:**
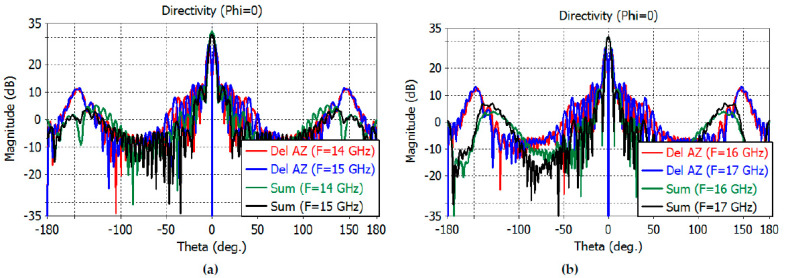
(**a**) *H*-plane directivity patterns of the reflector at 14–15 GHz (−180° to 180°). (**b**) *H*-plane directivity patterns of the reflector at 16–17 GHz (−180° to 180°).

**Figure 14 sensors-23-00118-f014:**
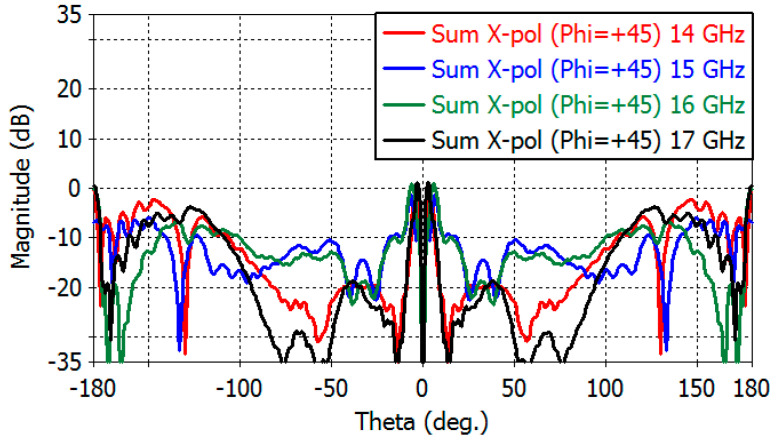
Diagonal-plane cross-polarization directivity of the sum channel (−180° to 180°).

**Figure 15 sensors-23-00118-f015:**
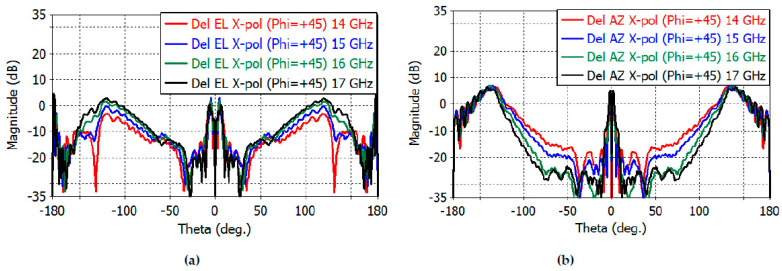
Diagonal-plane cross-polarization directivity of the elevation-difference channel (**a**) and of the azimuth-difference channel (**b**) (−180° to 180°).

**Table 1 sensors-23-00118-t001:** Dimensions of the feed and the truncated parabolic reflector.

**Parameter**	*d* (mm)	*a* (mm)	*b* (mm)	*g* (mm)
**Value**	65.0	16.0	8.50	3.50
**Parameter**	*K* (mm)	*L* (mm)	*C* (mm)	*E* (mm)
**Value**	36.5	4.00	8.00	1.00
**Parameter**	*ε_r_*	*F* (mm)		
**Value**	2.10	3.33		

**Table 2 sensors-23-00118-t002:** Dimensions of the truncated parabolic reflector.

**Parameter**	*F* (mm)	*D* (mm)	*θ*_1_ (deg.)	*θ*_2_ (deg.)	*L_t_* (mm)	*W_t_* (mm)
**Value**	259	500	51	23	452.20	213.34

**Table 3 sensors-23-00118-t003:** Comparative Analysis.

Ref.	*f* (GHz)	*aG*_SUM_ (dBi)	*G*_DIFF_ (dBi)	*D* (mm)	BW*_E_* (deg.)	BW*_H_* (deg.)	SLL (dB)
[9]	34.5–35.5	32	28	140	4.24	4.24	−25
[11]	W-band	42.2	38.6	-	1.41	1.61	−35
[19]	9.2	34.4	-	1200 × 300	6.10	1.60	−22.7/−18.9
This work	14–17	30.9–32.1	28.1–29.9/26.4–27.8	452 × 213	2.6–3.2	4.6–5.2	−20.0 to −12.6/−20.0 to −18.8

## Data Availability

Not applicable.

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
