# Peer review of "Design of a Ku-Band Monopulse Antenna with a Truncated Reflector and an Open-Ended Waveguide Feed"

_sensors, 2022, doi:10.3390/s23010118_

Round 1

Reviewer 1 Report

1. how do you validate the results ? the results are not compared with measured results.

2. The results are not compared with existing results. Justify?

3. sufficient Papers are not referred in the literature survey

4. Rewrite the abstract and conclusion 

Author Response

Response to Reviewer 1 Comments

Point 1: how do you validate the results ? the results are not compared with measured results.

Response 1:

Upon the reviewer's comment, we have modified the manuscript as follows.

"The purpose of our paper is to present an idea not to show the result of development efforts. In this regard we have used CST Studio Suite 2022, a well-proven simulation tool whose accuracy has been confirmed in many published papers so that the simulation alone is enough to prove the proposed concept."

Point 2: The results are not compared with existing results. Justify?

Response 2:

Upon the reviewer's comment, we have compared this results with the existing results in the manuscript as follows.

"The table shows the comparative analysis between this work and existing results. It can be seen from Table 3 that this work achieved asymmetrical beam performance at 14-17 GHz with sharp beams at the elevation and spread out azimuth beams useful for applications such as projectile tracking radars, airport surveillance radars, and vehicle tracking radars."

Comparative Analysis

Ref.

f (GHz)

GSUM

(dBi)

GDIFF (dBi)

D (mm)

BWE (deg.)

BWH (deg.)

SLL (dB)

[9]

34.5-35.5

32

28

140

4.24

4.24

-25

[11]

W-band

42.2

38.6

-

1.41

1.61

-35

[19]

9.2

34.4

-

1200 × 300

6.10

1.60

-22.7/-18.9

This work

14-17

30.9-32.1

28.1-29.9/ 26.4-27.8

452 × 213

2.6-3.2 

4.6-5.2

−20.0 to −12.6/ −20.0 to −18.8

Point 3: sufficient Papers are not referred in the literature survey.

Response 3:

Upon the reviewer's comment, we have reffered more papers in the literature survey. Added references [14]-[19].

Point 4: Rewrite the abstract and conclusion

Response 4:

Upon the reviewer's comment, we have, modified the abstract: below are added to the abstract: “The design and analysis have been carried out using the commercial simulation tool CST Studio Suite 2022, and its accuracy has been confirmed in many published papers”.

Upon the reviewer's comment, we have, we have modified the conclusion.  

Reviewer 2 Report

This paper presents a design of a monopulse feed truncated parabolic reflector antenna for radar applications at Ku band. I think, paper is interesting. I would propose some changes as follows:

1.      Authors should stress novelty of their work in comparison with others.

2.      It would be highly desirable to compare the obtained results with the experimental outputs.

3.      Authors are missing some recent articles in the field such as 
Electric Field Distributions of the Fast and Slow Modes Propagated in the Open Rod SiC Waveguide, etc.

4.      Authors should justify the choice of the frequency range under consideration.

Author Response

Response to Reviewer 2 Comments

Point 1: Authors should stress novelty of their work in comparison with others.

Response 1:

In the Introduction, we have explained some of novelties. Upon the reviewer's comment, we have added the following.

" The feed is very simple and fed by coaxial probes. It is easier to implement a monopulse comparator in coaxial meidum than in waveguide form."

"We employed four rectangular waveguide open-ends which is compact and simple to realize."

Point 2: It would be highly desirable to compare the obtained results with the experimental outputs.

Response 2:

Upon the reviewer's comment we have modified the manuscript as follows.

"Essentially, the purpose of our paper is to present an idea not to show the result of development efforts. In this regard we have used CST Studio Suite 2022, a well-proven simulation tool whose accuracy has been confirmed in many published papers so that the simulation alone is enough to prove the proposed concept."

Point 3: Authors are missing some recent articles in the field such as

Electric Field Distributions of the Fast and Slow Modes Propagated in the Open Rod SiC Waveguide, etc. 

Response 3:

Has been added.

Point 4: Authors should justify the choice of the frequency range under consideration.

Response 4:

Upon the reviewer's comment, we have added the below to the manuscript:

"The design is made for a low-power Ku band frequency range radar system."
